# Solution-Blown Aligned Nanofiber Yarn and Its Application in Yarn-Shaped Supercapacitor

**DOI:** 10.3390/ma13173778

**Published:** 2020-08-26

**Authors:** Jingjing Yang, Zhaofei Mao, Ruiping Zheng, Hao Liu, Lei Shi

**Affiliations:** 1School of Textile Science and Engineering, Tiangong University, Tianjin 300387, China; 1830011047@tiangong.edu.cn (J.Y.); 1831015036@tiangong.edu.cn (Z.M.); rpingzheng@aliyun.com (R.Z.); liuhao@tiangong.edu.cn (H.L.); 2Institution of Smart Wearable Elect-textile, Tiangong University, Tianjin 300387, China

**Keywords:** yarn-shaped supercapacitor, solution blowing, nanofiber yarn, yarn electrode, carbon fiber bundles

## Abstract

Yarn-shaped supercapacitors with great flexibility are highly anticipated for smart wearable devices. Herein, a device for continuously producing oriented nanofiber yarn based on solution blowing was invented, which was important for the nanofiber yarn electrode to realize mass production. Further, the yarn-shaped supercapacitor was assembled by the yarn electrode with the polypyrrole (PPy) grown on aligned carbon fiber bundles@Polyacrylonitrile nanofibers (CFs@PAN NFs). Electrical conductivity and mechanical properties of the yarn electrode can be improved by the carbon fiber bundles. The specific surface area of the yarn electrode can be enlarged by PPy. The yarn-shaped supercapacitors assembled by the PVA/LiCl/H_3_PO_4_ gel electrolyte showed high areal specific capacitance of 353 mF cm^−2^ at a current density of 0.1 A g^−1^, and the energy density was 48 μWh cm^−2^ when the power density was 247 μW cm^−2^. The supercapacitors also exhibited terrific cycle stability (82% after 20,000 cycles). We also proved that this yarn-shaped supercapacitor could easily power up the light emitting diode. This yarn-shaped supercapacitor was meaningful for the development of the smart wearable devices, especially when combined with clothing or fabrics.

## 1. Introduction

Yarn-shaped supercapacitors are attracting more and more attention due to their wide application potential in smart wearable devices in recent years. This type of supercapacitors has the following characteristics: light weight, fast charge/discharge, splendid flexibility, safety, and so on [1,2,3,4,5]. Especially, they can be used for knittable and weavable energy storage devices [6,7]. The important component of yarn-shaped supercapacitors is the flexible yarn electrodes, and the yarn electrodes must be high capacitance and perfect cycle retention rate. There are two ways to improve the electrochemical performance. The one way is to enlarge the specific surface area, as we all know, the high specific surface area can be obtained by coated nanofibers [8]. Another way is using conducting polymers, like polyaniline (PANI), poly(3,4-ethylenedioxythiophene) (PEDOT), polypyrrole (PPy), whose theoretical capacitances are 2000 F g^−1^ [9], 480 F g^−1^ [10,11] and 210 F g^−1^ [10,12], respectively. Among these conducting polymers, PPy is considered to be very promising polymer for great environment stability, high electrical conductivity and easy to synthesize, especially suitable for bonding with nanofiber yarn. For instance, Sun et al. fabricate a yarn-shaped supercapacitor based on PPy@CNTs@urethane elastic fiber core spun yarns (UY) which shows an areal specific capacitance of 67 mF cm^−2^ [13]. Based on polypyrrole-coated stainless steel/cotton blended yarns, this all-solid-state yarn-shaped supercapacitor presents a great areal specific capacitance of 344 mF cm^−2^ when the current density is 0.6 mA cm^−2^ [14].

The most important step to make a yarn-shaped supercapacitor is to produce a suitable yarn. An ideal yarn electrode material is always oriented, especially, the yarn is used to coated PPy [15]. However, the oriented nanofiber yarn is hard to realize. Nowadays, oriented nanofiber yarn is commonly produced by electrospinning [16,17,18,19,20], but mass production still could not be realized. Nanofiber yarn can realize mass production by solution blowing with its high solution feed rate which has come to an anticipant to be studied deeply [21]. Our group has used this technique to produce the nanofiber yarn by the auxiliary device. Zhuang et al. replace the previous mesh-like collector in solution blown device with a rotary disk [21]. In the spinning process, a funnel-like nanofiber net is formed to collect nanofibers and finally twisted to align the nanofiber yarns. The nanofiber yarn was carbonized and used as electrode material. Jia et al. use a pair of parallel rods as collector to modify the solution blown device to prepare oriented PAN nanofiber yarns. After carbonization, carbon nanofiber yarns are obtained for electrode materials [22]. However, these two methods can not realize the mass production of oriented nanofiber yarns.

In order to solve this problem, we introduced a modified solution blowing process which can realize mass production to straightway fabricate oriented nanofiber yarn. We used carbon fiber bundles (CFs) as core and polyacrylonitrile nanofibers (PAN NFs) as shell to produce CFs@PAN NFs core spun yarn. According to the characteristics of the yarn, we reported a supercapacitor based on a CFs@PAN NFs@PPy electrode whereby polypyrrole (PPy) was grown on aligned CFs@PAN NFs by in-situ polymerization. The yarn-shaped supercapacitor was assembled by PVA/LiCl/H_3_PO_4_ gel electrolyte and the results showed that the yarn electrodes revealed high areal specific capacitance (353 mF cm^−2^), great energy density (48 μWh cm^−2^), outstanding power density (247 μW cm^−2^ ), and exhibited excellent cycle stability (82% after 20,000 cycles).

## 2. Materials and Methods

### 2.1. Materials

All chemical reagents were analytical grade (AR). Anhydrous ethanol, sodium sulfate (Na_2_SO_4_), Ferric chloride (FeCl_3_), p-toluenesulfonic acid, N,N-dimethylformamide (DMF), lithium chloride (LiCl), and phosphoric acid (H_3_PO_4_) were purchased from Tianjin Fengchuan Industrial Co., Ltd. (Tianjin, China); pyrrole (Py) monomer was purchased from Shanghai Kewang Industrial Co., Ltd. (Shanghai, China); and polyacrylonitrile (PAN) and polyvinyl butyral (PVA) were obtained from Sea Petrochemical Co., Ltd. and Shanghai Aladdin Industrial Co., Ltd. (Shanghai, China), respectively. Carbon fiber bundles (6K) was purchased by Toray Industries, Inc. (Tokyo, Japan).

### 2.2. Preparation of Carbon Fiber Bundles @ Polyacrylonitrile Nanofiers (CFs@PAN NFs)

As shown in Figure 1, the CFs@PAN NFs yarns were produced by the novel solution blowing device. The PAN NFs coated CFs were fabricated as follows: 6 wt%, 8 wt%, 10 wt%, 12 wt% and 15 wt% PAN in N, N-dimethylformamide (DMF) was put in the nozzle which was connected with high air flow. Carbon fiber bundles passed through the funnel and pulled down by the rotate collector. Solution blown nanofibers from nozzle gathered on the carbon fiber bundles superficially, the rotate round table was used to twist the aligned nanofiber bundles and then form oriented nanofiber yarn. The spinning solution feeding rate, air pressure supplied to the nozzle, and collector distance were 10 mL h^−1^, 0.4 Bar, and 30 cm, respectively.

### 2.3. Preparation of Carbon Fiber Bundles@Polyacrylonitrile Nanofibers@polypyrrole (CFs@PAN NFs@PPy)

The CFs@PAN NFs@PPy electrode was obtained by the in-situ polymerization. PPy was grown on CFs@PAN NFs via a chemical deposition method for 0.5 h, 1 h, 2 h or 4 h in a solution of 0.6 mL PPy, 2.2 g FeCl_3_, 2.6 g P-toluene sulfonic acid and 60 mL distilled water in bath. The achieved electrode was made aligned nanofiber yarn into CFs@PAN NFs@PPy. Before chemical deposition, pyrrole was diluted in 30 mL distilled water. Figure 2 showed the fabrication process of the CFs@PAN NFs@PPy electrodes. The carbon fiber bundles was used as a current collector and it also can improve the conductivity and mechanical properties of PAN nanofiber yarns.

### 2.4. Assembly of the Yarn-Shaped Supercapacitor

Yarn-shaped supercapacitors were based on CFs@PAN NFs@PPy electrodes. The first step was to make the PVA/LiCl/H_3_PO_4_ gel electrolyte: 6 g PVA was introduced into 60 mL DI water and the solution stirred at 90 °C for 2 h, after which the 6 g H_3_PO_4_, 6 g LiCl was mingled. In the second step, all-solid-state yarn-shaped supercapacitors assembly by gel electrolyte, two yarn electrodes were twisted and dipped in PVA/LiCl/H_3_PO_4_ gel electrolyte. At last, the assembled yarn-shaped supercapacitors were dried at normal temperature whole day (Figure 2).

### 2.5. Characterization

Fourier transform infrared spectroscopy (FTIR) was measured by (Spectrum Two, PerkinElmer, company, Waltham, MA, USA). Scanning electron microscopy (SEM) was achieved from Phenom LE (Phenom China Ltd., Shanghai, China) and SU4800 (HITACHI Ltd., Tokyo, Japan). The specific surface area of samples was tested by an automatic gas adsorption analyzer (Quantachrome Instruments Ltd., Shanghai, China) and calculated using Brunauer–Emmett–Teller (BET) method.

### 2.6. Measurements

The electrochemical properties of these yarn-shaped supercapacitors can be measured by the electrochemical workstation (CHI 660E, CH Instruments Ins., Shanghai, China). The cyclic voltammetry (CV), galvanostatic charge/discharge (GCD), and EC impedance spectra (EIS) of these yarn electrodes were tested by two/three-electrode system. The cycle retention rate can be obtained by the CT 2001A battery program controlling test system (China-Land Co., Ltd., Wuhan, China).

The areal specific capacitances (*C_A_*) can be calculated by the GCD curves.
(1)$$C_{A}=\frac{I\cdot t}{U\cdot A}$$
where *I*, *t*, and *U* are the response current, discharge time, and potential window, respectively. The formula A = πDL is applied to figure out the superficial area of a yarn electrode. The equation of *E_A_* = *C_A_U*^2^/2 is used to count the areal energy density (*E_A_*), in which *C_A_* is areal specific capacitance of the whole device. For instance, areal power density (*P_A_*) can be computed by *P_A_* = *E_A_/t*.

## 3. Resullts and Discussion

SEM characterization was applied to explore the morphology of the nanofiber yarn, which was showed in Figure 3. Figure 3 summarized the surface morphology of nanofiber yarns at different concentration. Figure 3a–e showed the morphology of CFs@PAN NFs yarn with different PAN concentrations (6 wt%, 8 wt%, 10 wt%, 12 wt%, and 15 wt%) at the same magnifications to find the optimal solution concentration. The surface morphology of the PAN NFs could be clearly seen by magnifying the core spun yarn by 8000 times as shown in Figure 3h. Among these, Figure 3a showed the nanofiber yarn with 6 wt% PAN with a structure that was disorderly, especially the carbon fiber bundles was still visible. 8 wt% PAN nanofiber yarn was already forming but unevenness in terms fineness could be seen in Figure 3b,f. With the increasing of solution concentration, the coverage of PAN NFs was improved and then achieved the best orientation and uniformity (Figure 3g). The 10 wt% PAN NFs became obvious orientation, uniform fineness and smooth surface. It could continuously produce nanofiber yarn without droplets with 10 wt% PAN (Figure 3c). The amount of nanofiber was enough to completely cover the carbon fiber bundles, and the droplets were reduced but became larger. PAN NFs was not stretched completely before the solvent volatiles because the viscous resistance was increased when the solution concentration was up to 12 wt% and 15 wt% as shown in Figure 3d,e. As shown in Figure 3f, according to the standard deviation and uniformity of data comparison showed that the evenness and fineness of nanofiber yarn was the best when the PAN concentration was 10 wt%. The photo of the nanofiber yarn was more intuitive (Figure 3g). As shown in Figure 3i and Figure 4b, the diameters of CFs@10 wt%-PAN NFs and CFs@10 wt%-PAN NFs@1 h-PPy were respectively 0.856 mm and 0.867 mm, and the nanofiber yarns were well aligned and twisted without microscopically identifiable droplets were shown in Figure 3h. PAN NFs shell with about 50 μm thick was densely coated on CFs (Figure 3i). At a low magnification (Figure 3i), the surface of CFs@PAN NFs@PPy was similar to that of CFs@PAN NFs.

As shown in Figure 4, the nanofibers coated could create a rough surface structure, which can create a high surface area for the deposition of PPy. The BET surface area of the CFs@10 wt%-PAN NFs and CFs@10 wt%-PAN NFs@1 h-PPy was about 6.397 m^2^ g^−1^ and 15.955 m^2^ g^−1^, respectively, which was much higher than that of the carbon fiber bundles. Additionally, PAN NFs were relatively short, the incontinuity may disturb ion inter-nanofiber transport, then led to bad electrochemical property, so CFs were used as core yarn to solve this question [23]. For the CFs@PAN NFs@PPy yarn electrode, the surface of the carbon fiber bundles was roundly covered by the PAN nanofibers and the CFs@PAN NFs yarn was coated by PPy nanoparticles. This oriented structure of nanofiber yarn was projected to promote the ion transport and to increase the interface between active materials and electrolytes [15].

Morphology and FTIR of the yarn electrode were tested. Figure 4a showed the morphology of CFs@10 wt%-PAN NFs@PPy electrodes with different polymerization time at the same magnifications. The in-situ polymerization of PPy on CFs@10 wt%-PAN NFs was proceed by soaking the nanofiber yarn in aqueous solution with a specific pyrrole concentration. During the polymerization process, the nanofiber yarn blackened, and the PPy was first taken shape on the nanofibers and then reached onto the carbon fiber bundles. And the polymerization time was seted at 0.5 h, 1 h, 2 h, or 4 h, respectively. At high magnification, both the carbon fiber bundles and PAN NFs were evenly coated with uniform PPy nanopractiles, in which the homogeneity of PPy nanoparticles depended on the pyrrole polymerization time during in-situ polymerization (Figure 4b,c). PPy with 0.5 h polymerization time was found to disorderly grow on PAN NFs (Figure 4a (I)). Further increasing the polymerization time to 1 h, PPy evenly dispersed on the yarn surface (Figure 4a (II)). When the pyrrole polymerization time was 2 h, the agglomeration was beginning (Figure 4a (III)). As shown in Figure 4a (IV), with the polymerization time increasing, the agglomeration became more and more serious. Therefore, based on the previous conclusion, the structure of CFs@10 wt%-PAN NFs@1 h-PPy certainly was conducive to the ion diffusion and electron transfer.

FTIR was tested to affirm that the PPy had successful grown on PAN NFs yarns (Figure 4d). The A curve showed two pyrrole ring stretching peaks at 1470 cm^−1^ and 1560 cm^−1^, with C–H and C–N stretching peaks at 1330 cm^−1^ and 1040 cm^−1^ indicating that the PPy had been successfully grafted onto the PAN nanofiber yarn. Moreover, the strong absorption peak at 1180 cm^−1^ was also caused by PPy doping. In Figure 4d, the appearance of C≡N in the B curve proved that the PAN NFs coated on the CFs successfully. One vibration at 2930 cm^−1^ in the CFs@PAN NFs spectrum was attributed to the tensile vibration of the C-H, while the remaining bending vibration peaks at 1450 cm^−1^, 1350 cm^−1^, and 1380 cm^−1^ were also C–H.

The electrochemical properties of these yarn electrodes were evaluated by conducting the CV and GCD tested by three-electrode system in the electrolyte of 1 M Na_2_SO_4_ (Figure 5).

Figure 5a showed CV curves of CFs@10 wt%-PAN NFs@PPy with different polymerization time at 50 mV s^−1^ scanning rate. The shape of these curves was rectangle-like. It may be connected with introducing carbon fiber bundles, as we all know, CFs also can be used as active materials for electrochemical reaction [20]. Compared to other electrodes (Figure 5a), CFs@10 wt%-PAN NFs@1 h-PPy possessed larger areas at the scan rate of 50 mV s^−1^. Increasing the polymerization time resulted in changed electrochemical performance. Lower or longer polymerization time (0.5 h or 2 h and 4 h) led to smaller CV areas with inapparent redox peaks (Figure 5a). However, when the polymerization time was 1 h, the CV curves symbolized the faradaic pseudo-capacitance, thus the yarn electrode had higher specific capacitance. CV curves showed in Figure 5b, CFs@PAN NFs@1 h-PPy with different concentration(6 wt%, 8 wt%, 10 wt%, 12 wt%, and 15 wt%) of PAN at 20 mV s^−1^ scanning rate. Additionally, CFs@10 wt%-PAN NFs@1 h-PPy based electrode (Figure 5b) displayed much better electrochemical performance compared to other electrodes. This indicated that suitable yarn structure had a fantastic influence on the electrochemical property of the yarn electrode. The CFs@10 wt%-PAN NFs@1 h-PPy (Figure 5c) showed a nearly rectangular shape at different scan rate (2 mV s^−1^, 5 mV s^−1^, 10 mV s^−1^, 20 mV s^−1^, 50 mV s^−1^, 100 mV s^−1^), and the areal specific capacitance was 649 mF cm^−2^. Besides, these CV curves showed great symmetry suggested that the yarn electrode had terrific reversible oxidation-reduction reactions.

The GCD test was conducted to further research the electrochemical property. Theoretically, when the capacitance was caused by the electrical double layer, the GCD curves usually showed in regular triangle shape. As shown in Figure 5d–f, these GCD curves were displayed to have an almost regular triangle shape [24]. The quick charging action was nearly due to the electrical double layer capacitance between CFs and PAN NFs. A slow charging rate was mostly put down to pseudo-capacitance in PPy. Therefore, suitable polymerization time of Py leaded to longer charging time. Figure 5d showed GCD curves of the CFs@10 wt%-PAN NFs@PPy electrode with different polymerization time at the current density of 2 A g^−1^. As shown in Figure 5d, CFs@10 wt%-PAN NFs@1 h-PPy electrode had much longer discharge times (60.1 s) than other electrode at the same current density. The GCD curves of CFs@PAN NFs@1 h-PPy with different PAN concentration(6 wt%, 8 wt%, 10 wt%, 12 wt% and 15 wt%) were tested at 1 A g^−1^ current density as shown in Figure 5e. In Figure 5e, CFs@10 wt%-PAN NFs@1 h-PPy had a better electrochemical properties than other yarn electrode which indicated oriented nanofiber yarn had a great influence on electrode materials. In particular, CFs@10 wt%-PAN NFs@1 h-PPy electrode showed the longest discharge time (130 s when the current density was 2 A g^−1^) and biggest capacitance among these electrodes with different PAN concentration. Therefore, suitable concentration and special structure would resulted in much better electrochemical performance. Finally, we further studied the GCD of the CFs@10 wt%-PAN NFs@1 h-PPy. At different scan rates (0.1 A g^−1^, 0.2 A g^−1^, 0.5 A g^−1^, 1 A g^−1^, and 2 A g^−1^), the CFs@10 wt%-PAN NFs@1 h-PPy (Figure 5f) illustrated triangle-like shape and the GCD curves exposed the small resistance and quick electron transfer. Also, when at 0.1 A g^−1^ current density, the discharge time was the longest (1308 s) for CFs@10 wt%-PAN NFs@1 h-PPy. Among the left and right parts, these GCD curves showed excellent symmetry, which reveled good reversibility of charge/discharge reaction.

In order to further study the resistance of the yarn electrode, EIS tests of the yarn electrode were characterized at a frequency range of 0.01 Hz-100 kHz and the amplitude was 5 mV (Figure 5g–i). The equivalent series resistance and the charge transport resistance can be seen by the x-intercept of Nyquist plot [25,26]. The high-frequency region of the slope in Nyquist plot, it was on behalf of ion diffusion resistance between electrolyte and electrode interfaces [20]. Without obvious semicircle area in the Nyquist plot, it showed that the internal resistance between PAN NFs and CFs was low, and the electron inter-transport was good [16]. Furthermore, CFs@10 wt%-PAN NFs@1 h-PPy had the largest slope as illustrated in Figure 5g–i, revealing the smallest ion diffusion resistance (6.48 Ω). These consequences showed that the oriented structure should have ample region for ion storage and transport. The explaination for this phenomenon was that the different structure of the PAN NFs caused by different PAN concentration. When the surface area increased, the attachment point of polypyrrole was added, and the effective area of the contact between active material and electrolyte was improved, thus augmenting the exchange rate of electrolyte ions.

CFs@10 wt%-PAN NFs@1 h-PPy was used to assemble the yarn-shaped supercapacitor. Compared with common supercapacitors, all-solid-state yarn-shaped supercapacitors with great promise have the following advantages: great flexibility, light weight, high stability, and more safety [27,28]. So, we used gel electrolyte to assemble the yarn-shaped supercapacitor. In Figure 6, the electrochemical properties of yarn-shaped supercapacitors assembled by PVA/LiCl/H_3_PO_4_ gel electrolyte were roundly tested, these measurements included the cyclic voltammetry (CV), galvanostatic charge/discharge (GCD) and cycle retention rate. Figure 6a,b showed CV curves of yarn-shaped supercapacitor at scanning rates from 2 mV s^−1^ to 100 mV s^−1^ and GCD curves at current density from 0.1 A g^−1^ to 2 A g^−1^. The areal specific capacitance of the supercapacitor tested in PVA/LiCl/H_3_PO_4_ gel electrolytes was 353.26 mF cm^−2^. The stability of yarn-shaped supercapacitors was measured in this work, as well. A cycle stability test was arranged to figure out the cycle retention rate of the CFs@10 wt%-PAN NFs@1 h-PPy based yarn-shaped supercapacitors. Figure 6c showed that the areal specific capacitance still remained nearly 80% of its starting value after 20,000 cycles, manifesting its outstanding cycle retention rate. The cycle retention rate of the supercapacitor was about 100% at the 10,000th cycle, and 82% at the 20,000th cycle. The charge and discharge curves of the supercapacitor was showed at 19,996–20,000 cycles in the Figure 6c. The GCD curve of supercapacitor was nearly no change, and its shape was triangle. Table 1 showed that the areal specific capacitance, areal energy density and areal power density of CFs@10 wt%-PAN NFs@1 h-PPy yarn-shaped supercapacitors were much better than lately reported yarn-shaped supercapacitors based on PPy [13,14,15,29,30,31]. The areal energy density of the CFs@10 wt%-PAN NFs@1 h-PPy yarn-shaped supercapacitors was nearly eight times higher than the supercapacitor based on the electrode of the PPy@CNTs@urethane elastic fiber (6.13 μWh cm^−2^) [13], about one point five times higher than PPy/SS/cotton (36.2 μWh cm^−2^) [14], three times higher than PPy/BC [15], two times higher than PPy/CNT-ionic liquid/AuNP/carbon fiber (24.7 μWh cm^−2^) [29], five times higher than PPy/MnO_2_/rGO (9.2 μWh cm^−2^) [31]. This comparison showed that the CFs@10 wt%-PAN NFs@1 h-PPy electrode materials reported in this paper provided a very promising method to produce yarn electrodes for the components of a high-flexibility yarn-shaped supercapacitor.

Finally, two 8 cm long CFs@10 wt%-PAN NFs@1 h-PPy electrodes were used to assemble the yarn-shaped supercapacitor in PVA/LiCl/H_3_PO_4_ gel electrolyte and it can power up a red LED. To deeply test the flexibility of the yarn-shaped supercapacitor, the GCD curves were measured with different bending angles (Figure 7a–d). And the encapsulating material of the yarn-shaped supercapacitor was Eco-flex. From Figure 7e, it was found that the GCD curves of the yarn-shaped supercapacitor with different bending states did not change obviously. An 8-cm long yarn-shaped supercapacitor can brighten a light emitting diode for nearly two minutes while the voltage window was 2 V, and the series resistance was 15.2 Ω. Results displayed that the yarn-shaped supercapacitor based on CFs@10 wt%-PAN NFs@1 h-PPy was flexible enough to be used in smart wearable devices.

## 4. Conclusions

In this work, the modified solution blowing process with a funnel and a turntable as collector was a novel method to produce continuous aligned nanofiber yarn, and the nanofiber yarn performed well when it was used as a yarn electrode material for yarn-shaped supercapacitors. We successfully produced yarn-shaped supercapacitors based on CFs@10 wt%-PAN NFs@1 h-PPy electrodes. The carbon fiber bundles can improve the electrical conductivity and mechanical properties of the yarn-shaped supercapacitors. Moreover the supercapacitor based on CFs@10 wt%-PAN NFs@1 h-PPy electrodes presented high specific capacitance (353.26 mF cm^−2^ at the current density of 0.1 A g^−1^), great energy density (48 μWh cm^−2^ when the power density was 247 μW cm^−2^), and excellent cycling stability (82% after 20,000 cycles). Eventually, the yarn-shaped supercapacitors can easily lighten the light emitting diode, and this fact revealed that it had actual applications in wearable electronics and smart textiles. However, the gel electrolyte used in this paper was prone to dry. Thus, in the next work, we will try to invent a new gel electrolyte with improved water retention.

## Figures and Tables

**Figure 1 materials-13-03778-f001:**
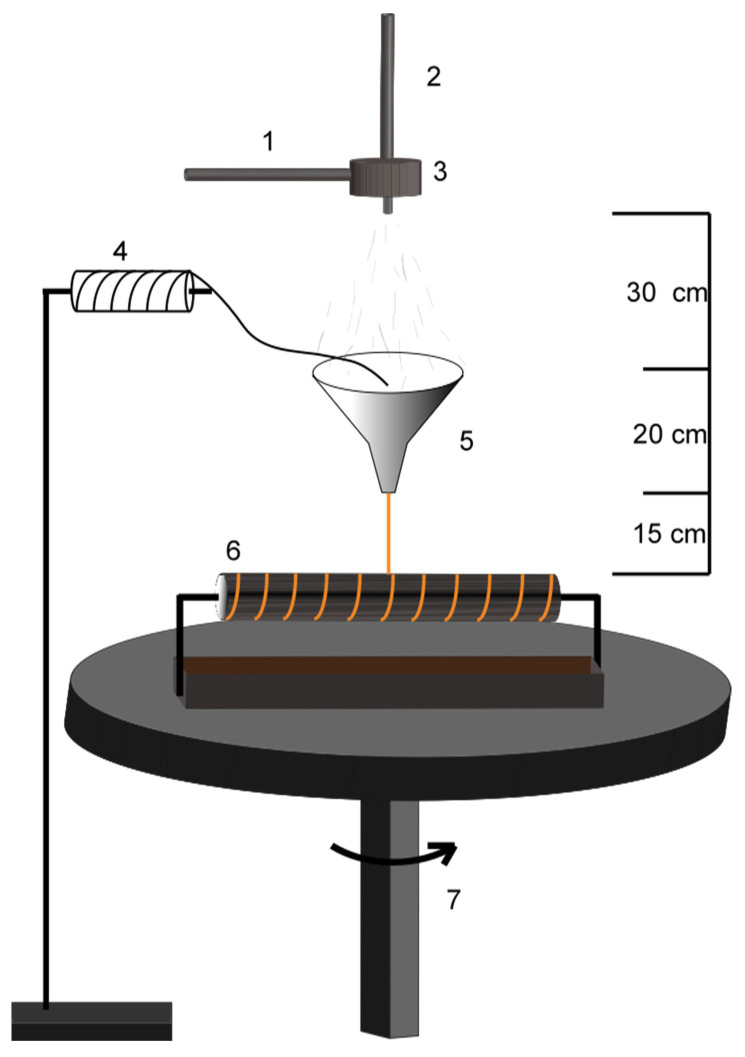
Schematic diagram of the Solution blowing process (1. solution; 2. air flow; 3. nozzle; 4. bobbin; 5. funnel; 6. rotate collector; 7. rotate round table).

**Figure 2 materials-13-03778-f002:**
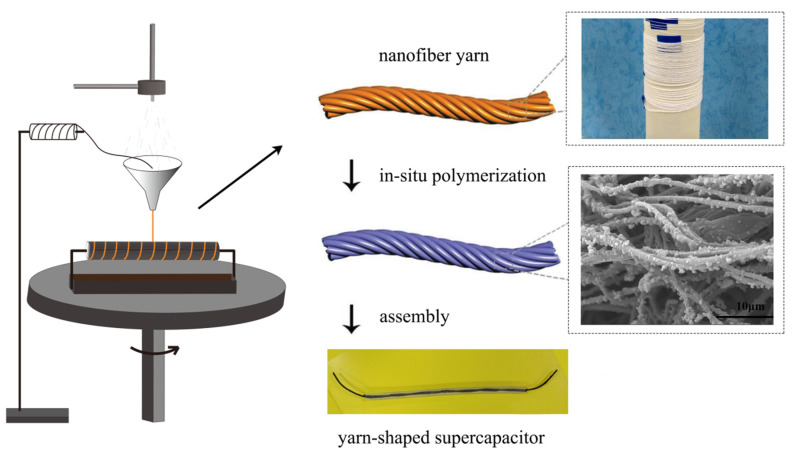
Process diagram of the supercapacitor based on CFs@PAN NFs@PPy.

**Figure 3 materials-13-03778-f003:**
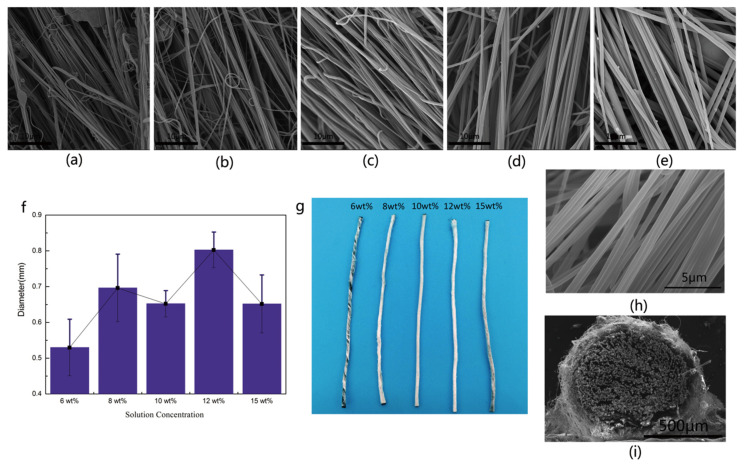
Surface morphology of (**a**) CFs@6 wt%-PAN NFs, (**b**) CFs@8 wt%-PAN NFs, (**c**) CFs@10 wt%-PAN NFs, (**d**) CFs@12 wt%-PAN NFs and (**e**) CFs@15 wt%-PAN NF, (**f**) diameter distribution and variance diagram of nanofiber yarn with different concentrations, (**g**) photo of the yarn, (**h**) morphology of nanofibers, (**i**) cross sectional morphology of CFs@10 wt%-PAN NFs.

**Figure 4 materials-13-03778-f004:**
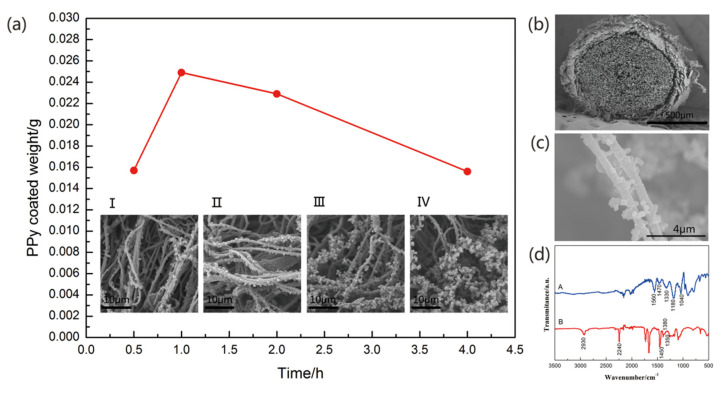
(**a**) PPy coated weight, and the morphology of (I) CFs@10 wt%-PAN NFs@0.5 h-PPy, (II) 10 wt%-PAN NFs@1 h-PPy, (III) 10 wt%-PAN NFs@2 h-PPy and (IV) 10 wt%-PAN NFs@4 h-PPy; (**b**) Cross sectional morphology of 10 wt%-PAN NFs@1 h-PPy; (**c**) morphology of PPy nanoparticles on the surface of the nanofiber; (**d**) FT-IR spectra.

**Figure 5 materials-13-03778-f005:**
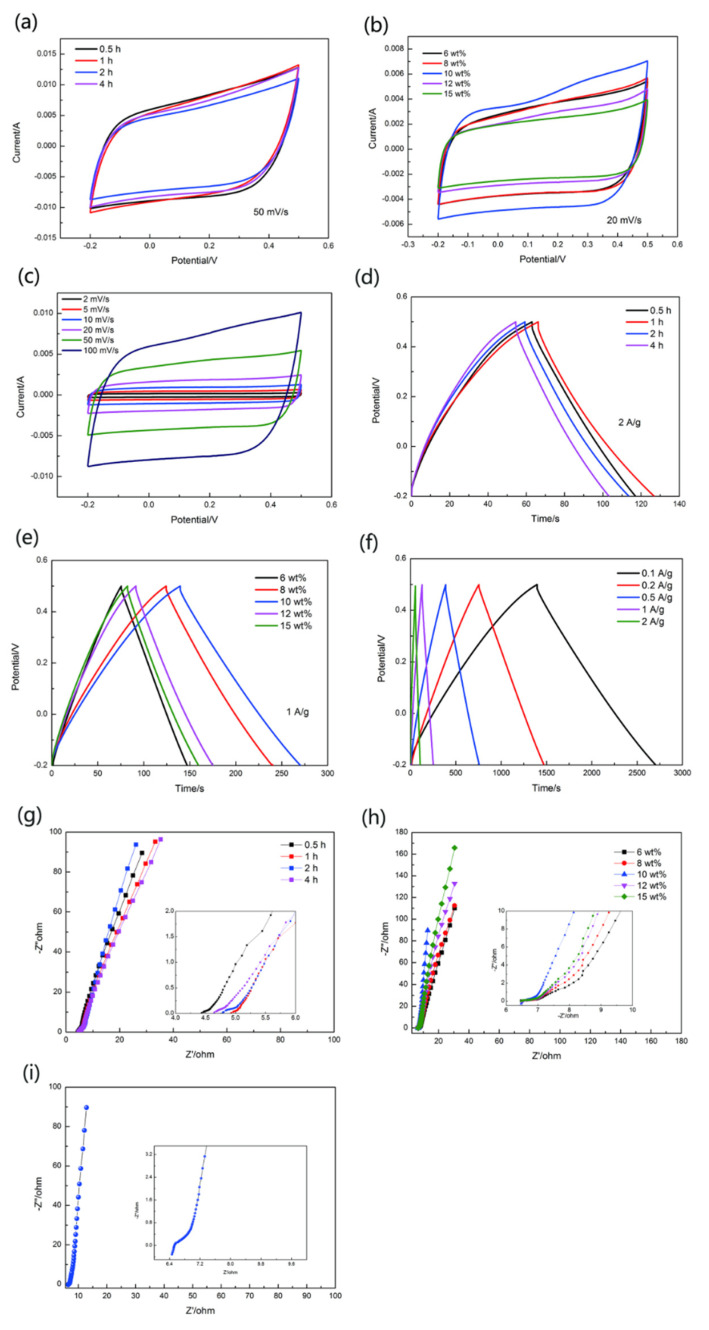
Electrochemical performance of yarn-shaped electrodes. (**a**) CV curves, (**d**) GCD curves, and (**g**) Nyquist plots of the CFs@10 wt%-PAN NFs@PPy with different desposited time by 0.5 h, 1 h, 2 h, 4 h. (**b**) CV curves, (**e**) GCD curves, and (**h**) Nyquistplots of CFs@PAN NFs@1 h-PPy with different concentrations(6 wt%, 8 wt%, 10 wt%, 12 wt% and 15 wt%). (**c**) CV curves, (**f**) GCD cvrves, and (**i**) Nyquistplots of CFs@10 wt%-PAN NFs@1 h-PPy.

**Figure 6 materials-13-03778-f006:**
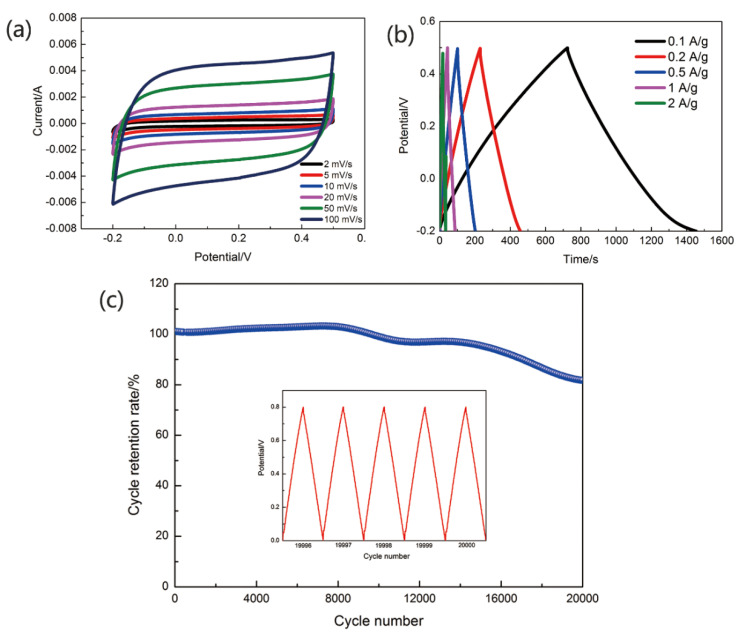
Electrochemical performance of yarn electrodes. (**a**) CV curves, (**b**) GCD curves, (**c**) the cycle retention rate of supercapacitors and charge-discharge curves of 19,996–20,000 cycle.

**Figure 7 materials-13-03778-f007:**
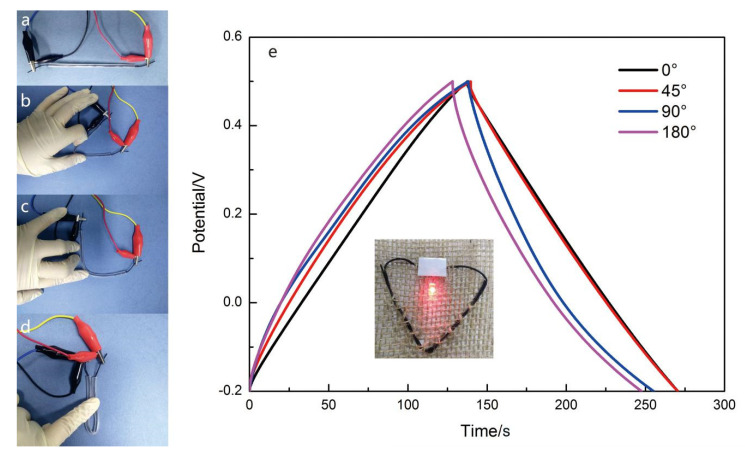
Images of the yarn supercapacitor under various bending state. (**a**) 0°, (**b**) 45°, (**c**) 90° and (**d**) 180°, (**e**) GCD curves of the yarn-shaped supercapacitor at 1 A g^−1^.

**Table 1 materials-13-03778-t001:** Comparison of PPy-based supercapacitor.

Electrode Materials	Areal Capacitance	Energy Density	Power Density	References
	(mF cm^−2^)	(μWh cm^−2^)	(μW cm^−2^)	
PPy@CNTs@urethane elastic fiber	67	6.13	133	[13]
PPy/SS/cotton	344	36.2	135	[14]
PPy/BC	76.6	16.9	10.9	[15]
PPy/CNT-ionic liquid/AuNP/carbon fiber	38.49	24.7	3520	[29]
PEDOT:PSS-PPy 50 wt% SSF/cotton yarn	1368.2	160	-	[30]
PPy/MnO_2_/rGO	103	9.2	1330	[31]
CFs@10 wt%-PAN NFs@1 h-PPy	353	48	247	This work

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
