# Peer review of "Solution-Blown Aligned Nanofiber Yarn and Its Application in Yarn-Shaped Supercapacitor"

_materials, 2020, doi:10.3390/ma13173778_

Round 1
Reviewer 1 Report
The authors reported the yarn-based supercapacitor whose efficiency is very high compared with the previous studies.
However, It is very hard to read because of lots of mistakes. For example, mismatch the explanation in the text and Figures.
In this light, there is no novelty for the publication in Materials.
Author Response
Manuscript number: materials-903514
MS Type: Article
Title: "Solution-blown aligned nanofiber yarn and its application in yarn-shaped supercapacitor"
Correspondence Author: Shi Lei
Email address: 1830011047@tiangong.edu.cn
Dear reviewer:
Thank you very much for your attention and comments on our paper materials-903514.
We have studied reviewers’ comments carefully and have made revision which marked in red in the paper according to your kind advice and other reviewers’ detailed suggestions.We sincerely hope this manuscript will be finally acceptable to be published on Materials. We have studied comments carefully and have made correction which we hope meet with approval. The main corrections in the paper and the responds to the your comments are as the following:
Response to Reviewer1 Comments
Point 1: The authors reported the yarn-based supercapacitor whose efficiency is very high compared with the previous studies. However, It is very hard to read because of lots of mistakes. For example, mismatch the explanation in the text and Figures. In this light, there is no novelty for the publication in Materials.
Response 1: Thanks for your comments on our paper, your recognition of our study gives us more motivation to improve our defective work. We have revised our paper roundly according to your comments, in particular, the problem of language and mismatch the explanation in the text and Figures. Considering the your suggestion, we have done these changes in the text:
Line 103, the "Figure 7" was changed to "Figure 2", Figure 2 is the schematic illustration of the fabrication of the CFs@PAN NFs@PPy based supercapacitor.
Line 139, the "Figure 1(f)" was changed to "Figure 3(f)", Figure 3(f) is the standard deviation and uniformity of the diameter of nanofiber yarn.
Line 187, the "Figure 2(d)" was changed to "Figure 4(d)", Figure 4(d) is the FT-IR spectra of the CFs@PAN NFs and CFs@PAN NFs@PPy.
Line 192-196, "Figure 5. Electrochemical performance of yarn-shaped electrodes. (a) CV curves, (d) GCD curves, and (g) Nyquist plots of the CFs@10 wt%-PAN NFs@PPy with different desposited time by 0.5 h, 1 h, 2 h, 4 h. (b) CV curves, (e) GCD curves, and (h) Nyquistplots of CFs@PAN NFs@1 h-PPy with different concentrations(6 wt%, 8 wt%, 10 wt%, 12 wt% and 15 wt%). (c) CV curves, (f) GCD cvrves, and (i) Nyquistplots of CFs@10 wt%-PAN NFs@1h-PPy." Among these, Figure 5(b-c, h, f) was mismatch in the previous text.
Line 207, Figure 5(b) in the previous text was 3b, which was a clerical error and had been modified in the new paper.
Line 146, 257 and 268, the tense error has been corrected, "is" has changed to "was".
Special thanks to you for your good comments.
If you have any questions or you think this article need to improve, please do not hesitate to contact us.

Reviewer 2 Report
In this paper, the authors present a study about the device for continuously producing oriented nanofiber yarn based on solution blowing, which was important for nanofiber yarn electrode to realized mass production. The authors also improved the supercapacitor for power-up LEDs. This article is clear, concise, and suitable for the scope of the journal. Several suggestions are supplied:
- Suggest improving the resolution of Fig.1.
- Suggest increase the font in Fig.6 (a) (b)
- Suggest give more information in the sentence about the comparison in table 1.
Author Response
Manuscript number: materials-903514
MS Type: Article
Title: "Solution-blown aligned nanofiber yarn and its application in yarn-shaped supercapacitor"
Correspondence Author: Shi Lei
Email address: 1830011047@tiangong.edu.cn
Dear reviewer:
Thank you very much for your attention and comments on our paper materials-903514.
We have studied reviewers’ comments carefully and have made revision which marked in red in the paper according to your kind advice and other reviewers’ detailed suggestions. We sincerely hope this manuscript will be finally acceptable to be published on Materials. We have studied comments carefully and have made correction which we hope meet with approval. The main corrections in the paper and the responds to the reviewer’s comments are as the following:
Response to Reviewer 2 Comments
Point 1: Suggest improving the resolution of Fig.1.
Response 1: Thanks for the referee’s kind suggestion. According to your advice, we have replaced the Figure 1 with a clearer picture, and the following is the corrected Figure 1.
Point 2: Suggest increase the font in Fig.6 (a) (b)
Response 2: We are very sorry for our negligence of this detail. We have resized the font in Figure 6, and the revised Figure 6 is showed as follows:
Point 3: Suggest give more information in the sentence about the comparison in table 1.
Response 3: We have re-written this part according to the reviewer’s suggestion. And we added the following sentence at line 270-275: The areal energy density of the CFs@10 wt%-PAN NFs@1h-PPy yarn-shaped supercapacitors was nearly eight times higher than the supercapacitor based on the electrode of the PPy@CNTs@urethane elastic fiber (6.13 μW h cm-2) [13], about one point five times higher than PPy/SS/cotton (36.2 μW h cm-2) [14], three times higher than PPy/BC [15], two times higher than PPy/CNT-ionic liquid/AuNP/carbon fiber (24.7 μW h cm-2 ) [29], five times higher than PPy/MnO2/rGO (9.2 μW h cm-2) [31].
Special thanks to you for your good comments.
If you have any questions or you think this article need to improve, please do not hesitate to contact us.

Reviewer 3 Report
47 and 82. The orientation in the fiber is the way in which its crystalline and amorphous parts are arranged. What do the authors understand by the term "oriented nanofiber yarn"?
79. How did the authors obtain carbon nanofiber yarn CFs?
88-90. Was the wrapped on the coil CFs@PAN NFs yarn dipped in a PPy bath or, as shown in Figure 2, was the bath dosed through needles to the funnel and rotate collector?
- The description of Figure 5 is not complete
- Figure 5 d,e,f - what is the time to reach the steady state (what is the time constant)?
- Figure 5 g,h,i - what are the conditions for obtaining Nyquist charts?
- voltage amplitude
- frequency range
- Which Figure are the authors describing 3b or 5b?
Figure 7 does not show the knitted yarn so what did the authors mean by writing: knitted yarn-shaped? Besides, the yarn obtained is very thick (Fig. 7). In all likelihood, the yarn obtained will not be suitable for processing on weaving and knitting machines, so how would the authors use this highly complex arrangement (obtained yarn) in wearable electronics?
How long was the LED lamp illustrated in the picture lit and at what series resistance?
To what voltage was the received supercapacitor charged in the experiment with a shining lamp?
Author Response
Manuscript number: materials-903514
MS Type: Article
Title: "Solution-blown aligned nanofiber yarn and its application in yarn-shaped supercapacitor"
Correspondence Author: Shi Lei
Email address: 1830011047@tiangong.edu.cn
Dear reviewer:
Thank you very much for your attention and comments on our paper materials-903514.
We have studied reviewers’ comments carefully and have made revision which marked in red in the paper according to your kind advice and other reviewers’ detailed suggestions. We sincerely hope this manuscript will be finally acceptable to be published on Materials. We have studied comments carefully and have made correction which we hope meet with approval. The main corrections in the paper and the responds to the reviewer’s comments are as the following:
Response to Reviewer 3 Comments
Point 1: 47 and 82. The orientation in the fiber is the way in which its crystalline and amorphous parts are arranged. What do the authors understand by the term "oriented nanofiber yarn"?
Response 1: Orientation refers to the parallel arrangement of molecular chains in the direction of external forces. The phenomena of polymer orientation includes the orientation of molecular chains, segments and the preferred arrangement of crystal polymer chips in the direction of external forces. The nanofiber yarn described in this paper is uniaxial orientation. Uniaxial orientation is the main method to form yarns, which means that the molecular chain is oriented along one direction by applying an external force on one axis. The ordered arrangement is an important feature for the formation of oriented nanofiber yarns, which can be observed and verified by scanning electron microscopy. Orientation nanofiber yarns: firstly, the nanofibers must be nanoscale; secondly, from a macroscopic point of view, as a yarn, the yarn has a uniform twist; finally, from a microscopic point of view, the nanofibers are in an orderly and regular way in the same direction.
Point 2: How did the authors obtain carbon nanofiber yarn CFs?
Response 2: Carbon fiber bundles (6K) was purchased from Toray Industries,Inc.
Point 3: Was the wrapped on the coil CFs@PAN NFs yarn dipped in a PPy bath or, as shown in Figure 2, was the bath dosed through needles to the funnel and rotate collector?
Response 3: To form the CFs@PAN NFs@PPy electrode, CFs@PAN NFs yarn was dipped in a PPy bath with a suitable length as shown in Figure 2.
Point 4: The description of Figure 5 is not complete. Figure 5 d,e,f - what is the time to reach the steady state (what is the time constant)? Figure 5 g,h,i - what are the conditions for obtaining Nyquist charts?- voltage amplitude- frequency range
Response 4: Considering the Reviewer’s suggestion, we have completed the description of the Figure 5. Line 197-251, these sentences describe the CV, GCD and EIS of the yarn electrode.
As shown in Figure 5(d), CFs@10 wt%-PAN NFs@1h-PPy electrode had much longer discharge times (60.1 s) than other electrode at the same current density. As shown in Figure 5(e), CFs@10 wt%-PAN NFs@1 h-PPy electrode showed the longest discharge time (130 s when the current density was 2 A g-1) and biggest capacitance among these electrodes with different PAN concentration. In Figure 5(f), when at 0.1 A g-1 current density, the discharge time was the longest (1308 s) for CFs@10 wt%-PAN NFs@1h-PPy.
In order to further study the resistance of these yarn electrodes, EIS tests of the yarn electrode were characterized at a frequency range of 0.01 Hz-100 kHz and the amplitude was 5 mV (Figure 5(g-i)).
Point 5: Which Figure are the authors describing 3b or 5b?
Response 5: We are very sorry for our negligence to result in this mistake, Figure 5(b) at 207 in the previous text was 3b, which was a clerical error and had been modified in the new text.
Point 6: Figure 7 does not show the knitted yarn so what did the authors mean by writing: knitted yarn-shaped? Besides, the yarn obtained is very thick (Fig. 7). In all likelihood, the yarn obtained will not be suitable for processing on weaving and knitting machines, so how would the authors use this highly complex arrangement (obtained yarn) in wearable electronics?
Response 6: The use of the word "knitted" was inconsiderate and the word had been changed in the latest version. The following was the revised section 279-286: "Finally, two 8 cm long CFs@10 wt%-PAN NFs@1h-PPy electrodes were used to assembled yarn-shaped supercapacitor in PVA/LiCl/H3PO4 gel electrolyte and it can power up a red LED. To deeply test the flexibility of the yarn-shaped supercapacitor, the GCD curves were recorded for the yarn-shaped supercapacitor with different bending angles which encapsulating material was Eco-flex (Figure 7(a-d)). From Figure 7(e), it was found that the shapes of GCD curves for the yarn-shaped supercapacitor with different bending states did not change significantly. These results proved that the yarn-shaped supercapacitor based on CFs@10 wt%-PAN NFs@1h-PPy was flexible enough to be used in smart wearable devices."
The yarn-shaped supercapacitor and its application in wearable textiles, we have two ideas to realize it. The first, yarn-shaped supercapacitor is combined with common yarn to fabricate slub yarn with special texture style, then use the slub yarn to form a fabric and the fabric can be used in different garment (please forgive us not to display the samples, conditions are not allowed to make samples). The second, as shown in the picture, we tried to combine the supercapacitor with traditional Chinese embroidery to make this sample. The common yarn and the yarn-shaped supercapacitor were designed to make up the dragonfly pattern. The supercapacitor not only supplies power for the wearable textile, but also plays a decorative role.
Point 7: How long was the LED lamp illustrated in the picture lit and at what series resistance?
Response 7: We have added this part appropriately in the article according to the reviewer’s comments (285-287). An 8 cm long yarn-shaped supercapacitor can brighten a LED for nearly two minutes at 2 V, and the series resistance is 15.2 Ω.
Special thanks to you for your good comments.
If you have any questions or you think this article need to improve, please do not hesitate to contact us.

Round 2
Reviewer 1 Report
I recommend the publication to Materials as it is.